# O-Serogroups and Pathovirotypes of *Escherichia coli* Isolated from Post-Weaning Piglets Showing Diarrhoea and/or Oedema in South Korea

**DOI:** 10.3390/vetsci9010001

**Published:** 2021-12-21

**Authors:** Jae-Won Byun, Bo-Youn Moon, Kyung-Hyo Do, Kichan Lee, Hae-Yeong Lee, Won-Il Kim, ByungJae So, Wan-Kyu Lee

**Affiliations:** 1Animal Disease Diagnostic Division, Animal and Plant Quarantine Agency, Gimcheon 39660, Korea; qiamby@korea.kr (B.-Y.M.); noanoa33@korea.kr (K.L.); bjso@korea.kr (B.S.); 2College of Veterinary Medicine, Chungbuk National University, Cheongju 28644, Korea; pollic@naver.com (K.-H.D.); wklee@chungbuk.ac.kr (W.-K.L.); 3Optipharm Inc., Cheongju 28158, Korea; bahl1@optipharm.co.kr; 4College of Veterinary Medicine, Chonbuk National University, Iksan 54596, Korea; kwi0621@jbnu.ac.kr

**Keywords:** pathotype, *Escherichia coli*, O-serogroup, pathovirotype, post-weaning diarrhoea and oedema, piglets

## Abstract

This study aimed to determine the prevalence of several pathovirotypes and evaluate the association of haemolysis with the virotypes of pathogenic *E. coli* isolated from post-weaning piglets in South Korea from 2015 to 2019. We isolated 890 *E. coli* and tested for O-serogroups, virulence genes, haemolysis, and multilocus sequence typing. The predominant virotypes were STb:EAST1:AIDA-I, F18b:Stx2e:AIDA-I, F18:STa:STb:Stx2e, and eae:Paa in enterotoxigenic *E. coli* (ETEC), Shiga toxin-producing *E. coli* (STEC), ETEC/STEC, and enteropathogenic *E. coli* (EPEC), respectively. Regarding serogroups, O139, O149, O141, and O121 were mostly detected in F18:Stx2e:AIDA-I, F4:LT:STb:EAST1, F18:STa:STb, and F18:Stx2e:EAST1, respectively. There was a significant change in the frequency of the O141:F18ac:STa:STb (an increase from 1.6% to 10.1%) and O139:F18ab:Stx2e:AIDA-I (a decrease from 13.0% to 5.3%) virotypes in ETEC and STEC, respectively, from 2015 to 2019. The O141:F18ac:STa:STb virotype was mostly detected in the central area and was spreading to the southern area. The odds ratios between haemolysis and virotypes were 11.0, 6.25, and 8.57 in F18:STa:STb, F18:Stx2e:AIDA-I, and F4:LT:STb:EAST1, respectively. Our findings provide insights regarding the recent prevalence of pathogenic *E. coli* in South Korea and could be used for the development of vaccines for *E. coli* responsible for PWD and ED in post-weaning piglets.

## 1. Introduction

Colibacillosis in post-weaning piglets, characterised by post-weaning diarrhoea (PWD) and oedema disease (ED), has economically affected the pig industry worldwide given the high morbidity rate, growth retardation, and elevated treatment costs. Colibacillosis is caused by pathogenic *Escherichia* (*E.*) *coli* that have acquired virulence genes involved in attachment (fimbriae and non-fimbrial adhesins) and toxins (heat liable (LT), heat-stable (ST), enteroaggregative heat-stable toxin 1 (EAST1), and Shiga-like toxin (Stx)). Based on their virulence factors, *E. coli* have been categorised into enterotoxigenic *E. coli* (ETEC), Shiga toxin-producing *E. coli* (STEC), enteropathogenic *E. coli* (EPEC), and extraintestinal pathogenic *E. coli* (ExPEC) [1,2,3]. PWD and ED are mainly associated with ETEC and STEC. Upon ingestion, these bacteria attach to mucosal layers of the small intestine using an attachment apparatus (F4, F5, F6, F18, and F41) with/without non-fimbrial factors, adhesin involved in diffuse adherence (AIDA-I), and porcine-attaching effacing factor (Paa). After colonisation, they produce enterotoxins (LT and ST) and/or Stx2e, which causes diarrhoea and oedema [4]. EPEC possess the eae gene (intimin), which facilitates intimate attachment with the intestinal epithelium and the attaching and effacement lesion. Diarrhoea is mainly caused by malabsorption [5]. 

F4 (K88) and F18 fimbriae are strongly associated with PWD and ED. F4 fimbriae have three antigenic variants (ab, ac, and ad), with F4ac being the most common in ETEC [5]. On the other hand, there are two variants of F18 fimbria: F18ab and F18ac. F18ab is mostly detected in Stx2e-encoding STEC strains while F18ac is associated with the ETEC and ETEC/STEC strains. A new F18 variant (F18new) has recently been identified in Germany and Korea [6,7].

There have been efforts to diagnose and characterise pathogenic *E. coli*. Generally, routine colibacillosis diagnosis in piglets is often performed through isolation, haemolysis on blood agar, O and H serotyping, and determination of the virulence factors using polymerase chain reaction (PCR or multiplex PCR), with its virulence pattern being used for virotyping [5,8]. Recently, multilocus sequence typing (MLST) has been used to determine the epidemiologic relevance of pathogenic *E. coli* in humans and animals [9,10]. It has more discriminatory power than existing methods, including pulse-field gel electrophoresis (PFGE) and PCR-based genotyping [11,12].

Previous studies have described the association between haemolysis and certain virulence factors [1,5,13,14]. However, the diagnostic utility of haemolytic *E. coli* strains isolated from diarrheic samples remains unclear. 

Although there have been numerous efforts to detect and characterise the pathogenic *E. coli* involved in PWD and ED, there remains a need for systematic national data to facilitate prevention strategies and vaccine development for colibacillosis. 

This study aimed to conduct national surveillance for pathogenic *E. coli* isolated from piglets with suspected PWD and/or ED from 2015 to 2019, as well as to characterise their phenotypes and genotypes. Moreover, we aimed to determine the diagnostic and predictive utility of haemolysis for pathogenic *E. coli* involved in PWD and ED. These results could provide novel insights regarding the recent prevalence of pathogenic *E. coli* in South Korea and the vaccine development for *E. coli* involved in PWD and ED in post-weaning piglets.

## 2. Materials and Methods

### 2.1. Bacterial Isolates

Annually, approximately 1000 faecal and/or intestinal samples obtained from post-weaned piglets with diarrhoea and/or oedema are submitted to four Animal Disease Diagnostic Labs (national lab: Gimcheon, Korea, private labs: Iksan, Korea, Cheongju, Korea, and Osong, Korea). From the samples, 890 samples containing *E. coli* were identified from 850 farms from 2015 to 2019. Farm information regarding the submission date, farm location, and affected age were obtained using a request form. The farm regions were divided into central (454 farms: Gyeonggi, Gangwon, Chungbuk, and Chungnam provinces) and southern (396 farms: Gyeongbuk, Gyeongnam, Jeonbuk, and Jeonnam provinces) areas.

Samples were streaked onto MacConkey (BD, Franklin Lakes, NJ, USA) and blood agar (Asan, Korea), followed by aerobic incubation for 18 h at 37 °C. Distinctive colonies that showed nearly pure culture on MacConkey (BD, Franklin Lakes, NJ, USA) and blood agar were transferred onto blood agar. After incubation, haemolysis was determined in the blood agar and isolates were identified as *E. coli* using the VITEK II system (bioMérieux, Craponne, France). Finally, the isolates were frozen in 10% glycerol at −70 °C until further use.

### 2.2. Determination of Virulence Genes

Genomic DNA of 890 *E. coli* isolates was extracted by boiling at 100 °C for five minutes. After centrifugation at 8000× *g*, the supernatant underwent PCR and real-time PCR. PCRs were performed for the genes of adhesions (F4, F5, F6, F18, F41, AIDA-I, and Paa) and toxins (LT, STa, STb, EAST1, and Stx2e) [4]. Reference strains of *E. coli* were provided by Dr J.M. Fairbrother (reference laboratory of *E. coli*, Canada) and were employed as positive controls for PCR analysis. The strains included 7805 (F4:LT:STa:STb:EAST1:Paa), 6611 (Stx1:Stx2:eae:EAST1:Paa), 1033 (F18:AIDA), 2316 (F6:STa: STb:EAST1:Paa), 13316 (F5:F41:STa:Paa), and 3463 (negative control). Real-time PCR assays were performed to differentiate the F18 subtypes on a CFX real-time PCR system (Bio-Rad Laboratories, Hercules, CA, USA), as previously described [6]. Samples of *E. coli* ((107/86:F18ab and 2134P:F18ac) and (KEFS025: F18new, APQA)) were obtained from Dr E. Cox (Ghent University, Belgium). The reaction volume (20 μL) comprised 2× ExTaq Mix (Takara, Tokyo, Japan), 2 μM of each primer, and 3 μL of DNA template. After amplification, products were visualised using microelectrophoresis (QIAxcel, Qiagen, Germantown, MA, USA).

### 2.3. O-Serogroups

O-serogroups were determined through PCR, as previously described [15], as well as through slide agglutination using rabbit polyclonal antisera to 181 O antigens [16]. Briefly, each overnight LB broth (BD, Franklin Lake, NJ, USA) culture was pelleted through centrifugation at 8000× *g*. Subsequently, the pellet was homogenised and heated at 100 °C for 5 min. The centrifuged supernatant was used as a DNA template. Upon amplification of the Og44, Og153, Og117, Og13, Og89, Og186, and Og46 primers, the isolates were incubated in tryptic soy broth (BD, Franklin Lakes, NJ, USA) overnight at 37 °C and heated for 2.5 h at 100 °C. Moreover, autoagglutination was determined using a mixture of 0.85% sodium chloride (NaCl) before testing the O-serogroup.

### 2.4. Multilocus Sequence Typing (MLST)

All 50 F18:STa:STb isolates underwent MLST genotyping, as previously described [12]. The PCR amplification and sequencing of seven housekeeping genes (*adk, fumC, gyrB, icd, mdh, purA,* and *recA*) were performed based on the protocols specified on the *E. coli* MLST website (http://mlst.warwick.ac.uk/mlst/dbs/Ecoli, accessed on 17 January 2021). All the primer sequences of these seven housekeeping genes are available at http://mlst.warwick.ac.uk/mlst/dbs/Ecoli/documents/primersColi (accessed on 17 January 2021). Allele numbers for seven gene fragments of each isolate were obtained through comparison with the corresponding alleles available in the MLST *E. coli* database (http://mlst.warwick.ac.uk/mlst/dbs/Ecoli, accessed on 17 January 2021); additionally, the sequence type (ST) of each isolate was determined by combining the seven allelic profiles. MLST clusters were calculated using BioNumerics version 8.0 (Applied Maths, bioMérieux, St. Maartens-Latem, Belgium). The farm locations with the most prevalent STs were depicted on a map based on information regarding the latitude and longitude obtained from the Korean Animal Health Integrated System.

### 2.5. Statistical Analysis

Proportion differences between the haemolysis and virulence genes (virotypes) were tested using the χ^2^ test; however, Fisher’s exact test was used when the number of isolates in the contingency table was below five. Statistical significance was defined as a *p*-value < 0.05. Odds ratios (OR) and log ORs were determined as comparative measures. All statistical analyses were conducted using the Jamovi statistical package (version 1.6.15, The jamovi project). 

## 3. Results

### 3.1. The Pathotype and Virotype of E. coli

A total of 890 samples with *E. coli* were isolated from 850 farms and classified into the corresponding pathotypes. Despite the annual fluctuation in the frequency of pathotypes, the predominant pathotype was ETEC (56.5%), followed by STEC (19.2%), ETEC/STEC (8.7%), and EPEC (7.6%) from 2015 to 2019 (Figure 1). The EPEC frequency from 2017 to 2019 was 10%; however, only one isolate was detected from 2015 to 2016. The frequency of isolates that encoded none of the virulence genes decreased from 15% in 2015 to 4% in 2019. 

In ETEC, the major virotype was STb:EAST1:AIDA-I (18.3%), followed by F18:STa:STb (15.7%), STa:STb (10.2%), and F4:LT:STb:EAST1 (6.9%) from 2015 to 2019 (Table 1; Figure 2). Notably, the frequency of F18:STa:STb increased from 2% in 2015 to 13% in 2019. In STEC, the frequency of F18:Stx2e:AIDA-I decreased from 12% in 2015 to 8% in 2019. The most common virotype in ETEC/STEC was F18:STa:STb:Stx2e (16.8%). The most common virotypes in EPEC were eae:Paa (61.7%) and eae (38.2%).

### 3.2. O-Serogroup and Virotype

Table 1 shows the O-serogroups and virotypes of *E. coli* isolated from weaned piglets (Appendix A). This study determined 54 O-serogroups (61.2%, 548 of 890). Sixteen serogroups (O139, O141, O149, O8, O98, O138, O115, O121, O163, O157, O45, O119, O7, O186, and O182) accounted for 82.6% of all the serogroups. The predominant serogroup was O139 (10.6%), followed by O141 (7.6%) and O149 (6.9%). 

There were likely associations between certain O-serogroups and virotypes (Table 1 and Appendix A). In STEC:F18 and ETEC:F4, the O139, O141, O149, and O121 O-serogroups were linked to the F18:Stx2e:AIDA-I (95.3%, 61/64), F18:STa:STb (75.0%, 60/80), F4:LT:STb:EAST1 (71.4%, 25/35), and F18:Stx2e:EAST1 (71.4%, 15/21) virotypes, respectively. In ETEC:F-, the O98 (17.2%, 16/93), O117 (13.9%, 13/93), and O45 (7.5%, 7/93) O-serogroups were linked to the STb:EAST1:AIDA-I virotype. The O8 O-serogroup was linked to the LT:STb:EAST1 (61.5%, 8/13) and EAST1 (11.7%, 6/51) virotypes. The O9 (64.2%, 9/14) and O182 (13.4%, 7/52) O-serogroups were linked to the STa:STb:Paa and STa:STb virotypes, respectively. In STEC:F- and EPEC, the O119 (15.5%, 9/58) and O186 (21.4%, 9/42) O-serogroups were linked to Stx2e and eae:Paa, respectively. 

Figure 3 shows the frequency of serovirotypes encoding the F18 subtype (O141:F18ac:STa:STb, O139:F18ab:Stx2e:AIDA-I, and O121:F18new:Stx2e:EAST1). The frequency of O139:F18ab:Stx2e:AIDA-I decreased from 13.8% in 2015 to 5.8% in 2019, while that of O141:F18ac:STa:STb increased from 1.6% in 2015 to 10.1% in 2019. The frequency of O121:F18new:Stx2e:EAST1 was <4% from 2015 to 2019. Notably, the frequency of O141:F18ac:STa:STb was higher in the central area (9.2%) than in the southern area (3.6%) for 5 years; however, there was no between-region difference in the frequency in 2019.

### 3.3. MLST Genotypes of the F18:STa:STb

A total of 50 F18:STa:STb isolates were tested to identify the ST distributions (Figure 4). Among them, 45 isolates were genotyped and assigned to 10 STs. The remaining five isolates could not be assigned to any ST since they could not be identified in the ST database in PulsNet (https://pubmlst.org/, accessed on 17 January 2021). The isolates were mostly isolated from farms located in the densely populated central area (Figure 5). The most frequent sequence types were ST7323 (28), ST760 (7), ST155 (3), and ST3054 (2). These four ST isolates accounted for 80% of the 50 *E. coli* F18:STa:STb isolates and were mostly detected in the central area. The ST7323, ST3054, and ST760 were generally detected from 2017 to 2019; however, ST155 was only detected in the central area in 2019 (Figure 4 and Figure 5).

### 3.4. Association between Virotypes (Virulence Gene) and Haemolysis

Table 2 presents the association between virotypes (virulence genes) and haemolysis (Appendix A). Generally, isolates encoding the virulence genes of F4, F18, LT, STa, and Stx2e were associated with haemolysis (*p* < 0.01). However, the virulence genes of STb, EAST1, eae, Paa, and AIDA-I was negatively associated with haemolysis (*p* < 0.01). Further, the virulence genes of F18 (OR, 33.3), Stx2e (OR, 6.18), and LT (OR, 4.28) were strongly associated with haemolysis. 

Regarding virulence gene combinations, the ORs for F18:STa:STb and F18:Stx2e:AIDA-I were 11.00 and 6.25, respectively, which was relatively low compared with those of singular virulence genes. Contrastingly, the OR of F4:LT:STb:EAST1 was 8.57, which was higher than those of singular virulence genes (F4, 2.43; LT, 4.28; STb, 0.68).

## 4. Discussion

There have been numerous efforts to characterise pathogenic *E. coli* based on the O and H antigen serogroups, phenotypes (pathotypes and virotypes), and genotypes (e.g., through PFGE and MLST) [1,10,17]. There have been numerous studies to detect and characterise the pathogenic *E. coli* isolated from piglets showing PWD and ED in diagnostic labs. However, there is a need for comprehensive national surveillance to facilitate prevention strategies and vaccine development for colibacillosis. Accordingly, this study conducted national surveillance for pathogenic *E. coli* isolated from piglets with suspected PWD and/or ED in four diagnostic labs from 2015 to 2019; moreover, it characterised the phenotypes and genotypes. Additionally, we evaluated the diagnostic importance of haemolysis as a predictive factor for pathogenic *E. coli* in PWD and ED.

In this study, among 890 isolates, 16 O-serogroups (O139, O141, O149, O8, O98, O138, O115, O121, O163, O157, O45, O119, O7, O186, and O182) accounted for 82.6% of the O-serogroups detected from 2015 to 2019. The predominant serogroup was O139 (10.6%), followed by O141 (7.6%) and O149 (6.9%); Generally, a limited number of O-serogroups have been frequently detected in the pathogenic strains of *E. coli*, with spatial and temporal differences. In South Korea, various O-serogroups have been detected in PWD and ED. Kwon and others (1999) reported that O157 and O8 were dominant strains from 1995 to 1997 [18]. In the 2000s, the O149 was highly detected from 2008 to 2014 [7]. Our findings showed that O139 and O141 were the predominant O-serogroups from 2015 to 2016 and from 2017 to 2019, respectively. 

In this study, 57% (508 of 890) of the detected isolates were classified as ETEC. In ETEC, the major virotype was STb:EAST1:AIDA-I (10.4%), followed by F18:STa:STb (9.0%), STa:STb (5.8%), and F4:LT:STb:EAST1 (3.9%). The F18:STa:STb and F4:LT:STb:EAST1 virotypes were strongly associated with O141 and O149, respectively. There was a moderate relationship of STb:EAST1:AIDA-I with O98 and O115 and of STa:STb with O149 and O182. STEC isolates accounted for 19.2% (171 of 890) of the detected isolates. The most prevalent virotype was F18:Stx2e:AIDA-I (37.4%), followed by Stx2e (33.9%); moreover, they were mostly identified from ED. The F18:Stx2e:AIDA-I was strongly associated with O139; contrastingly, Stx2e could not be virotyped in most cases. Notably, the frequency of STEC (F18ab:Stx2e:AIDA-I) decreased from 2015 to 2019 while that of ETEC (F18ac:STa:STb) and EPEC (eae:Paa and eae) has increased since 2017 (Figure 1 and Figure 2). This suggested changes in the frequency of the major pathovirotypes over 5 years. 

The underlying reason for the shift of prevalent virotypes isolated from pathogenic *E. coli* since 2017 remains unclear. However, using vaccines and feed additive antibiotics for diarrhoea and oedema could partially have influenced the observed shifts even if environmental factors, including feeds, temperature, and hygiene, are predisposing factors for PWD and ED [7]. The decrease of the frequency of ETEC (F4:LT:STb) and STEC (F18:Stx2e:AIDA-I) would have been associated with the use of vaccines for pili and enterotoxin (F4 and LT) since the 1990s and approval of the vaccine (Stx2e subunit) for ED in 2019, respectively; however, there were no differences in the frequency of STEC (Stx2e) from 2015 to 2019 [7].

ETEC expressing F4:LT:STb, F4: LT:STb:EAST1, F18:STa/STb/Stx2e toxins was the dominant pathotypes causing PWD [4]. In Europe, ETEC was frequently isolated in pig farms with PWD and accounted for 36.1% (F4-ETEC) and 18.2% (F18-ETEC) of the outbreaks. Of the F4-ETECs, F4:LT:STb, F4:LT:STa:STb, and F4:STa:STb comprised 27.5%, 15.7%, and 10.1%, respectively. Among the F18-ETECs, F18:STa:STb and F18:Sta:STb:Stx2e comprised 9.0%, respectively [19].

ETEC-expressing fimbriae (mainly F4 and F18) and toxins have generally been detected; however, fimbriae negative virotypes (STb:EAST1:AIDA-I, STa:STb, and eae) have been detected in PWD [1]. Among them, the F4:LT:STb virotype is among the most prevalent in the USA [4], Europe [19], and Vietnam [13]. The F18:STa:STb is commonly detected in PWD; further, it is associated with O138 and O141 [1,8]. The role of F4- and F18-negative virotypes in PWD remains unclear. Moreover, ETEC isolates of the STb or STb:EAST-1 virotypes from weaned pigs may produce a non-fimbrial apparatus (AIDA-I) [20,21], which was originally detected in *E. coli* isolates from humans with diarrhoea [1]. Ngeleka and others (2003) reported that 20.5% of enterotoxigenic *E. coli* were STb: EAST1:AIDA-I or STb:AIDA-I, which can induce diarrhoea in an experimental infection using newborn pigs. EPEC is considered a PWD-related pathotype and does not encode any virulence factor of classic ETEC strains [22]. 

The F18ab and F18ac subtypes are closely associated with STEC and ETEC strains, respectively, which cause ED and PWD [23]. F18ab was common in O139-STEC encoding Stx2e and AIDA-I; moreover, F18ac was associated with O141-ETEC encoding STa and STb. In this study, F18new was associated with O121-STEC encoding Stx2e and EAST1. F18new variant isolates were associated with the ETEC, rather than the STEC, pathotype based on clinical and antigenic sequence differences [6,7].

MLST has been used in epidemiologic studies on cases or outbreaks attributable to pathogenic *E. coli* from human and animals [10,24]. Although three different MLST approaches have been used in the public database, the current study mostly used Achtman methods with seven-locus sequencing. The most frequent sequence types of F18:STa:STb were ST7323 (28), ST760 (7), ST155 (3), and ST3054 (2). These four ST isolates accounted for 80% of the 50 F18:STa:STb isolates, with most being isolated from the central area. The ST7323, ST3054, and ST 760 were generally detected from 2017 to 2019. Contrastingly, ST155 was newly detected in the central area in 2019, with subsequent spreading to other areas.

The relationship between haemolysis and the pathotype has been evaluated with respect to being a diagnostic criterion. In this study, isolates encoding the virulence genes of F4, F18, LT, STa, and Stx2e were associated with haemolysis (*p* < 0.01). ETEC isolated from PWD cases are mostly haemolytic (ETEC F4:LT:STb:EAST1, 9% or F18:Stx2e:AIDA-I, 7.2%); however, non-haemolytic strains can be observed. Luppi and others (2016) reported that 97.6% of ETEC cases were haemolytic. The remaining 2.4% of non-haemolytic ETEC isolates, which underwent consistent tests for haemolytic activity, were recovered in France, Italy, and Germany; moreover, they shared similar virotypes (F4, STa, and STb). Some O-serogroups (O138, O139, and O141) in ED are strongly associated with haemolysis [8]. However, a previous study reported that haemolysin is not an essential virulence factor in a challenge experiment involving a mutant missing haemolysin [25]. In our study, F18 and F4 were strongly associated with haemolysis in PWD and ED isolates. However, the ORs of the strains encoding eae, Paa, EAST1, and AIDA-I were < 1, which indicates a negative relationship between virulence genes and haemolysis. In virulence gene combinations, the ORs of the F18:STa:STb and F18:Stx2e:AIDA-I isolates were slightly lower than those for other F18-positive virotypes. However, the OR value for F4:LT:STb:EAST1 was higher than those for the isolates singularly encoding each gene (F4, LT, STb, and EAST1). This is consistent with previous findings regarding F4- and F18-expressing strains [1,19]. However, isolates encoding enterotoxin (STb, EAST1) and non-fimbrial genes (eae, Paa, and AIDA-I) were non-haemolytic. Therefore, haemolytic colonies are likely to be associated with F4- and F18-positive ETEC and STEC.

## 5. Conclusions

Our findings indicated that the STb:EAST1:AIDA-I, F18:Stx2e:AIDA-I, and eae:Paa virotypes were the most prevalent in ETEC, STEC, and EPEC isolated in piglets with PWD and/or ED. Notably, from 2015 to 2019, there was a decrease and increase in the frequency of the F18:Stx2e:AIDA-I and F18:STa:STb virotype, respectively. F18- and F4-positive ETEC and STEC were more likely to be associated with haemolysis in PWD and ED isolates. Our data provide insights regarding the recent prevalence of pathogenic *E. coli* in South Korea and could facilitate vaccine development in *E. coli* responsible for PWD and ED in post-weaning piglets.

## Figures and Tables

**Figure 1 vetsci-09-00001-f001:**
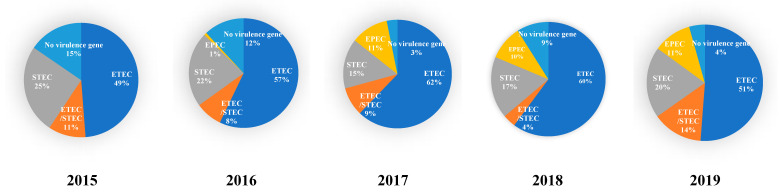
Pathotypes of *E. coli* isolated from the post-weaning piglets showing diarrhoea and oedema for 5 years (2015 to 2019).

**Figure 2 vetsci-09-00001-f002:**
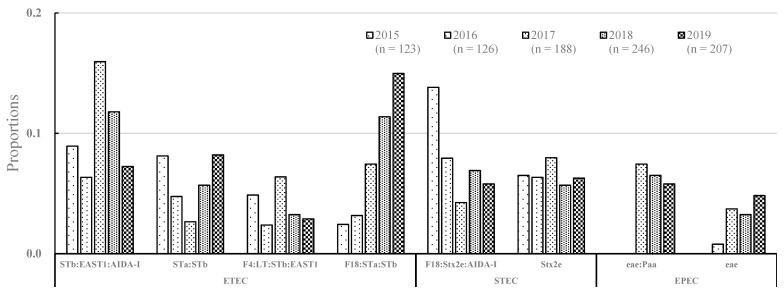
Major pathovirotype of *E. coli* isolated from the post-weaning piglets over 5 years (2015 to 2019).

**Figure 3 vetsci-09-00001-f003:**
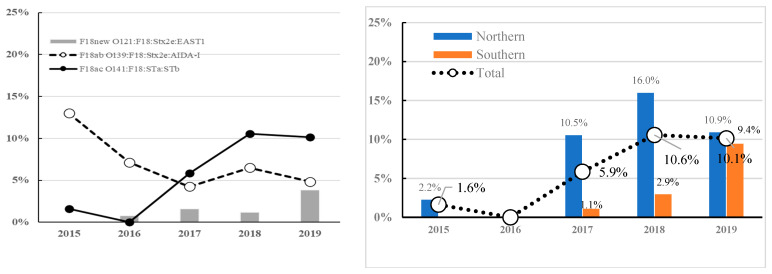
Change of F18 positive *E. coli* (O141:F18ac:STa:STb, O139: F18ab:Stx2e:AIDA-I and O121:F18new:Stx2e:EAST1) and regional frequency of dominant serovirotype (O141:F18ac:STa:STb) from 2015 to 2019.

**Figure 4 vetsci-09-00001-f004:**
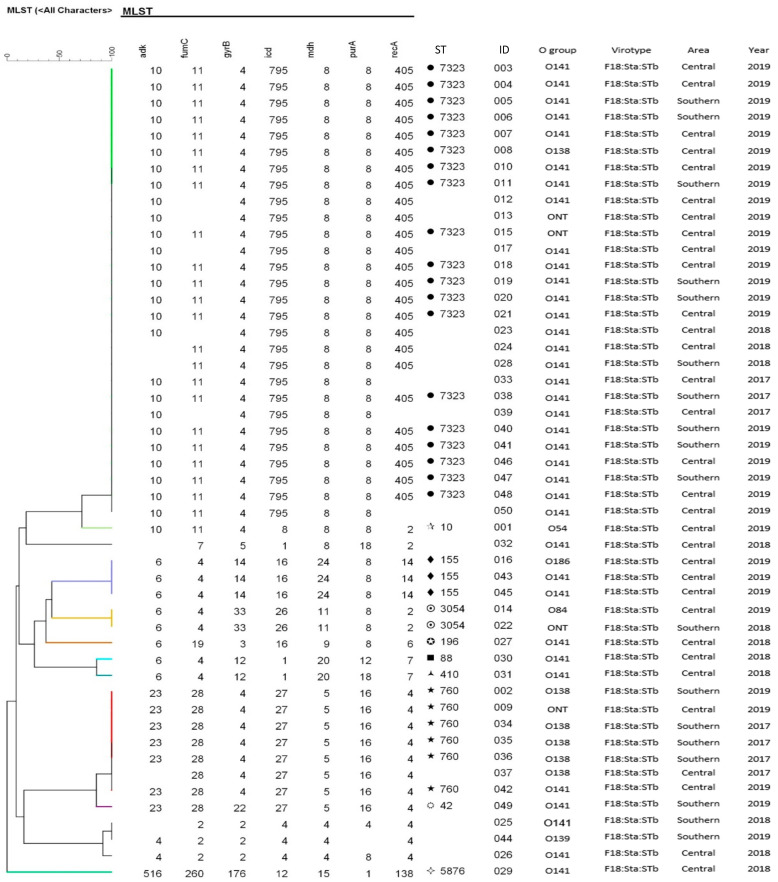
Maximum likelihood tree of sequence type (ST) of the F18:STa:STb detected in post-weaning piglets with O-serogroup, region detected, and year.

**Figure 5 vetsci-09-00001-f005:**
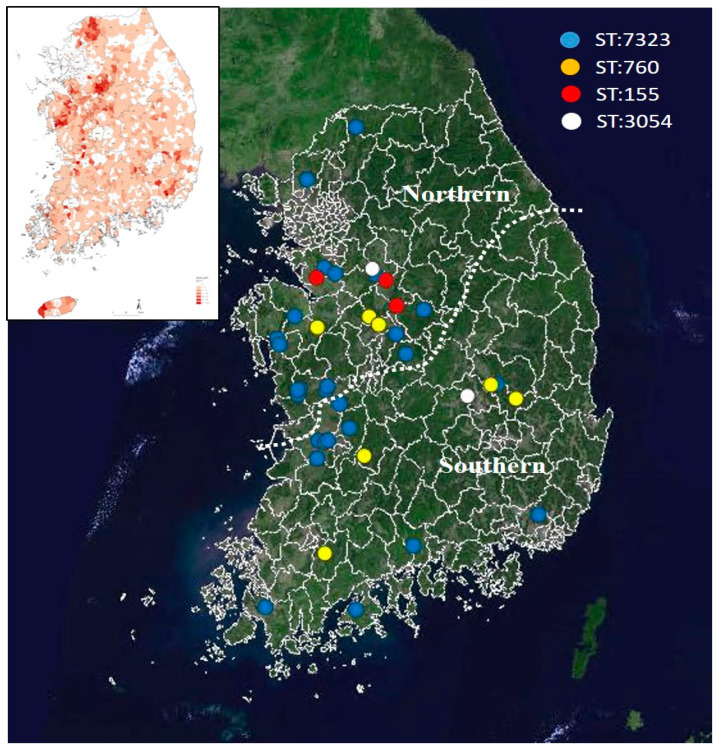
Regional distribution of F18:STa:STb virotype sequence type (ST7323, ST760, ST155, and ST3054) on the map. The density of all pig farms in South Korea.

**Table 1 vetsci-09-00001-t001:** O-serogroup and pathovirotypes of *E. coli* detected in post-weaning piglets.

Pathotype	Virotype	O-Serogroup																		Total
		O139	O149	O141	O8	O98	O9	O138	O115	O121	O163	O157	O45	O119	O7	O186	O182	Others *	ONT **	
ETEC:F-	STb:EAST1: AIDA-I	0	2	0	1	16	0	5	13	0	0	3	7	0	2	0	0	19	25	93
	STa:STb	0	6	2	2	0	3	3	1	0	0	0	0	1	0	0	7	8	19	52
	EAST1	0	1	0	6	0	0	0	0	0	0	2	0	1	4	0	0	11	26	51
	STa:STb:Paa	0	0	0	1	0	9	1	0	0	0	0	0	0	0	0	0	0	3	14
	LT:STb:EAST1	0	2	0	7	0	0	0	0	0	0	0	0	0	0	0	0	1	3	13
	LT:STb:EAST1:AIDA-I	0	0	0	1	0	0	0	0	0	1	0	0	0	2	0	0	0	9	13
	STb	0	0	0	1	0	0	0	1	0	0	0	0	0	0	0	0	1	7	10
	Other virotypes	0	3	0	1	0	0	0	2	0	2	0	0	0	0	0	0	0	19	27
	Subtotal	0	14	2	20	16	12	9	17	0	3	5	7	2	8	0	7	40	111	273
ETEC:F4	F4:LT:STb: EAST1	0	25	0	2	1	0	0	0	0	1	5	0	0	0	0	0	0	1	35
	F4:STa:STb	1	1	0	2	0	4	0	0	0	0	0	0	0	1	0	0	0	9	18
	F4:LT:STb: EAST1:Paa	0	15	0	0	0	0	0	0	0	0	0	0	0	0	0	0	0	1	16
	F4	0	0	0	0	1	0	0	0	0	1	0	0	0	0	0	0	1	4	7
	Other virotypes	0	2	0	0	0	0	0	0	0	0	1	0	0	0	0	0	1	7	11
	Subtotal	1	43	0	4	2	4	0	0	0	2	6	0	0	1	0	0	2	22	87
ETEC:F18	F18:STa:STb	1	0	60	0	1	1	4	0	0	2	0	0	0	0	1	0	3	7	80
	F18:EAST1	0	0	1	0	0	0	0	0	2	1	0	1	0	0	0	2	3	5	15
	F18:LT:STb: EAST1	0	1	0	0	0	0	0	0	0	0	4	0	0	0	0	0	2	5	12
	F18:Sta	2	0	0	0	0	0	1	0	0	3	0	0	0	0	0	0	0	5	11
	Other virotypes	5	1	2	1	3	0	0	0	0	2	0	3	0	2	0	0	5	6	30
	Subtotal	8	2	63	1	4	1	5	0	2	8	4	4	0	2	1	2	13	28	148
STEC:F-	Stx2e	3	1	0	0	3	1	1	1	0	0	0	0	9	0	0	0	9	30	58
	Stx2e:Paa	0	0	0	0	0	1	0	0	0	0	0	0	0	0	0	0	0	1	2
	Subtotal	3	1	0	0	3	2	1	1	0	0	0	0	9	0	0	0	9	31	60
STEC:F4	F4:Stx2e	0	0	0	0	0	0	0	0	0	0	0	0	0	0	0	0	0	7	7
	Subtotal	0	0	0	0	0	0	0	0	0	0	0	0	0	0	0	0	0	7	7
STEC:F18	F18:Stx2e:AIDA-I	61	1	2	0	0	0	0	0	0	0	0	0	0	0	0	0	0	0	64
	F18:Stx2e: EAST1	2	0	0	0	0	0	1	0	15	1	0	0	0	0	0	0	0	2	21
	F18:Stx2e	3	0	0	0	1	0	0	0	1	1	0	0	0	0	0	0	1	1	8
	F18:Stx2e:AIDA-I:Paa	3	0	0	0	0	0	0	0	0	0	0	0	0	0	0	0	0	2	5
	F18:Stx2e:eae	1	0	0	0	0	0	0	0	0	0	0	0	0	0	0	0	0	0	1
	Subtotal	70	1	2	0	1	0	1	0	16	2	0	0	0	0	0	0	1	5	99
ETEC/ STEC:F-	STa:STb:Stx2e	1	0	0	1	0	0	1	0	0	0	0	0	0	0	0	0	3	2	8
	STa:Stx2e	0	0	0	1	0	0	0	0	0	0	0	0	0	0	0	0	0	3	4
	Other virotypes	1	0	0	0	1	0	0	1	0	0	0	0	0	0	0	0	0	1	4
	Subtotal	2	0	0	2	1	0	1	1	0	0	0	0	0	0	0	0	3	6	16
ETEC/ STEC:F4	F4:STa:STb:Stx2e	0	0	0	1	0	0	1	0	0	0	0	0	0	0	0	0	1	4	7
	F4:LT:Stx2e	0	0	0	0	0	0	0	0	0	0	0	0	0	0	0	0	0	1	1
	F4:STa:Stx2e	0	0	0	0	0	0	0	0	0	0	0	0	0	0	0	0	1	0	1
	Subtotal	0	0	0	1	0	0	1	0	0	0	0	0	0	0	0	0	2	5	9
ETEC/ STEC:F18	F18:STa:STb:Stx2e	4	0	0	0	0	1	3	0	0	0	0	0	0	0	0	0	0	5	13
	F18:LT:Stx2e	0	0	0	1	0	0	1	0	0	0	0	0	0	0	0	0	0	10	12
	F18:STa:Stx2e	1	0	0	0	0	0	0	0	1	0	0	0	0	0	0	0	1	7	10
	Other virotypes	3	0	0	0	0	2	0	0	0	0	0	0	0	0	0	0	0	12	17
	Subtotal	8	0	0	1	0	3	4	0	1	0	0	0	0	0	0	0	1	34	52
EPEC	eae	0	0	0	0	0	0	0	0	0	0	0	0	0	0	0	0	8	18	26
	eae:Paa	0	0	0	0	2	0	0	0	0	0	0	1	0	0	9	0	12	18	42
	Subtotal	0	0	0	0	2	0	0	0	0	0	0	1	0	0	9	0	20	36	68
	No virulence gene	2	0	1	2	0	1	1	0	0	0	0	0	0	0	0	0	4	60	71
	Total	94	61	68	31	29	23	23	19	19	15	15	12	11	11	10	9	95	345	890

* Others (No. of strains): O15(8), O2(8), O14(7), O91(7), O132(6), O76(6), O84(4), O86(4), O101(2), O107(2), O108(3), O109(3), O49(3), O147(3), O111(2), O145(2), O153(2), O101(1), O112ab(1), O112ac(1), O128(1), O133(1), O154(1), O159(1), O17(1), O173(1), O174(1), O23(1), O48(1), O51(1), O54(1), O56(1), O63(1), O80(1), O87(1). ** ONT: Non-typable.

**Table 2 vetsci-09-00001-t002:** Association between virotype and haemolysis of *E. coli* detected in post-weaning piglets.

Virulence Gene	Virotype	No. of Isolates	Haemolysis *		χ^2^-Value	*p*-Value	Log Odds Ratio	Odds Ratio
			O	X				
	F4:LT:STb:EAST1	35	31	4	-	<0.001	2.15	8.57
	F18:STa:STb	80	72	8	58.8	<0.001	2.40	11.00
	F18:Stx2e:AIDA-I	64	54	10	34.3	<0.001	1.83	6.25
	Stx2e	58	40	18	9.80	0.002	0.89	2.43
	STb:EAST1:AIDA-I	93	2	91	-	<0.001	−4.00	0.01
	STa:STb	52	5	47	-	<0.001	−2.30	0.10
	EAST1	51	1	50	-	<0.001	−3.99	0.01
	eae	26	1	25	-	<0.001	−3.24	0.03
	Eae:Paa	42	1	41	-	<0.001	−3.77	0.02
	No virulence gene	71	29	48	2.10	0.147	−0.36	0.69
F4		103	68	35	13.3	<0.001	0.78	2.20
F18		303	280	23	345	<0.001	3.51	33.3
LT		142	109	33	51.7	<0.001	1.44	4.23
STa		260	153	107	14.0	<0.001	0.55	1.74
STb		430	190	240	7.66	0.005	−0.38	0.68
Stx2e		241	192	49	124	<0.001	1.87	6.46
EAST1		297	109	188	27.4	<0.001	−0.75	0.46
eae		72	4	68	-	<0.001	−2.95	0.05
Paa		91	23	68	23.0	<0.001	−1.16	0.31
AIDA-I		194	69	125	18.2	<0.001	−0.70	0.49

*: **O**, hemolysis; **X**, non-hemolysis.

## Data Availability

The data are available upon request from the authors.

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
