# Peer review of "O-Serogroups and Pathovirotypes of Escherichia coli Isolated from Post-Weaning Piglets Showing Diarrhoea and/or Oedema in South Korea"

_vetsci, 2021, doi:10.3390/vetsci9010001_

Round 1

Reviewer 1 Report

The study described in the manuscript aimed to conduct South Korea surveillance for pathogenic E. coli isolated from piglets with suspected PWD and/or ED from 2015 to 2019, as well as to characterize their phenotypes and genotypes. Generally, contents of all sections are appropriate and adequate. Materials and methods used in the study are adequately described. Results are generally well described and presented in the manuscript as well as discussion which is comprehensive. Conclusions were justified by the obtained results and correspond to the aim of the study. But some comments to the study appear and are listed below.

General comments

  • I understood the idea of reporting altogether pathovirotypes for post-weaning piglets but concerning the totally different clinical course of PWD and ED it will be additional value of the manuscript to show some of the results with breakdown for strains of coli causing PWD and/or ED.
  • I the discussion in my opinion lack of comparison of results to other countries reports is one of the weakness of the manuscript.

Specific comments

  • Figure 1. is illegible in terms of descriptions directly on wheel-chart.
  • On the Figure 4. titles of columns 8 and 9 are also illegible.
  • Line 205: Data with percentage of hemolytic strains will be useful especially if you have cited that kind of data in the discussion (Lines 293-295).
  • Is it possible to avoid repetition of results in the discussion section? (Lines 232-236 or 243-248)
  • The regionalization you have made divide South Korea for Southern and Central part, which in fact is the Northern part of South Korea.

Author Response

The manuscript has been changed according to recommendations 

Reviewer 2 Report

The authors provided an elaborate and interesting study to investigate the prevalence of several pathovirotypes and evaluate the association of haemolysis with the virotypes of pathogenic E. coli isolated from post-weaning piglets with diarrhea and/or oedema in South Korea from 2015 to 2019. And the study indicated that the STb:EAST1:AIDA-I, F18:Stx2e:AIDA-I and eae:Paa virotypes were the most prevalent in the ETEC, STEC, and EPEC isolated from the diarrheal piglets. However, there are some issues mainly in the Discussion section need to be solved by the authors.

  1. It is correct that the authors conduct chi-square test to analyze the proportion differences, but please explain why identify P < 0.01 as significant.
  2. The first paragraph in the Discuss section mostly repeats the part in the Introduction section.
  3. The author should discuss the results and provide the scientific hypothesis but not simply present them in the Discussion section, for instance, why the predominant O serogroups from 2015 to 2016 and from 2017 to 2019 were O139 and O141 compared to previous predominant O serogroups before 2015?
  4. Indeed the authors took numerous efforts to compile the database from different farms in the whole South Korea within 5 years, but the authors should analyze the current observations compared to others countries considering many factors such as use of antibiotics (difference between EU and US), feed formula (difference between corn-soybean style and barley-wheat style), season and so on. Otherwise, I really concern the representability of this study.
  5. The English written need to be improved.

Author Response

All comments has been revised according to reviewer recommendations

Reviewer 3 Report

In this research, the Authors determined the prevalence of different pathvirotypes and evaluate association of haemolysis with pathogenic E. coli virotypes isolated from post-weaning piglets in South Korea from 2015 to 2019. They isolated 890 E. coli and tested serogroup O, virulence 
genes, hemolysis and typing of multilocus sequences. The results obtained  provide insights into the recent prevalence of pathogenic E. coli in South Korea and could be useful to develop a vaccine  in E. coli responsible for PWD and ED in post-weaning piglets. The manuscript is well organized, the methodology used is correct and the results are interesting especially for the geographic area (South Korea) where the study was performed, but the data obtained could be useful for the development of vaccines for E. coli responsible for PWD and ED in post-weaning piglets.

Author Response

Thank you for your comments

Round 2

Reviewer 1 Report

I have no further comments.